



*PDO-driven interdecadal variability of snowfall over the Karakoram and Western Himalaya*
**Authors: Priya Bharati[1], Pranab Deb[1], Kieran M. R. Hunt[2,3]**
**1 CORAL, Indian Institute of Technology Kharagpur, Kharagpur, India**
**2 Department of Meteorology, University of Reading, Reading, UK**
**3 National Centre for Atmospheric Science, University of Reading, UK**
***Correspondence to**: pranab@coral.iitkgp.ac.in*





## Abstract:

Our study reveals that the negative phase of the Pacific Decadal Oscillation (PDO-) leads to
increased winter (DJF) snowfall in the Karakoram and Western Himalayas (KH) from 1940 to
2022. Interdecadal variations in DJF snowfall during the PDO- are attributed to deep convection
and adiabatic cooling near the tropopause in both the northwest Pacific and KH region.
Additionally, a wave-like pattern characterized by a trough (anomalous cyclone) north of KH and a
ridge (anomalous Tibetan Plateau anticyclone) east of KH in the upper atmosphere, along the
northward shift of the DJF Subtropical Jet (STJ) was observed. A strong positive correlation
between DJF STJ strength and DJF snowfall in KH as well as a significant negative correlation
between DJF STJ strength and DJF PDO, suggests a wave response over KH to the direct forcing
over the northwest Pacific Ocean. The intensified STJ across KH results in higher frequency of
Western disturbances, leading to anomalous moisture convergence and increased DJF precipitation
in the region during the PDO-. These findings hold significant implications for the decadal
predictability of winter snowfall in KH by the various phases of PDO.

## 1) Introduction:

Glaciers in the Karakoram and Western Himalaya (KH) exhibit unique stability compared to other
alpine glaciers (known as the 'Karakoram Anomaly'; Hewitt, 2005; Kaab et al., 2012; Gardelle et
al., 2013; Kapnick et al., 2014; Forsythe et al., 2017; de Kok et al., 2018; Farinotti et al., 2020;
HIMAP, 2020). Winter snowfall plays a significant role in preserving the local snowpack and
sustaining the glacial mass balance at higher elevations (Tahir et al., 2011; Bolch et al., 2012;
Ridley et al., 2013; Cannon et al., 2015; Dimri et al., 2015), and controls almost 60% of the
variability in glacier mass balance in the KH region (Kumar et al., 2019). The decline in average
and minimum summer temperatures, along with significant increases in winter, summer, and annual
precipitation, have been proposed as crucial factors influencing the stable glacier budget of the KH
in recent decades (Archer and Fowler, 2006; Forsythe et al., 2017).
The KH receives around 50% of its annual precipitation as snowfall from western disturbances
(WDs) (Lang and Barros, 2004; Barros et al., 2006; Bookhagen and Burbank, 2010; Hunt et al.,
2024). Furthermore, WDs account for more than 65% of all winter snowfall and nearly 53% of total
winter precipitation in the KH (Javed et al., 2022). However, using a less conservative method,
Midhuna et al. (2020) found that WDs account for about 80% of winter precipitation in KH. WDs



are upper level troughs in the subtropical westerly jet (STJ), which grow via baroclinic instability (Norris et al., 2015; Cannon et al., 2017; Hunt et al., 2018). Strong WDs are associated with deep uplift to the east of their centre and drive moist lower-tropospheric southwesterlies from the Arabian Sea (Dimri and Dash, 2012; Hunt et al., 2018), resulting in heavy precipitation along the foothills and mountains of KH region (Baudouin et al., 2020). The snowfall from WDs in the KH is heavily influenced by the complex topography of the region, as well as by synoptic and mesoscale factors (Cannon et al., 2015; Norris et al., 2015, 2017, 2018). Subsequent snowmelt in the following spring and summer seasons and associated runoff serve as major sources of downstream river flow and provide relief from drought to populations that are vulnerable to water stress (Bolch et al., 2012; Hewitt et al., 2014; Rana et al., 2019; Pritchard et al., 2019).

However, the main climatic drivers affecting seasonal precipitation, and hence glacial mass balance in the region are only partially understood (Cannon et al., 2015).WD activity during winter season over the KH has been reported to be influenced by several global climate forcings such as North Atlantic Oscillation/Arctic Oscillation (Yadav et al., 2009; Syed et al., 2010; Filippi et al., 2014; Basu et al., 2017; Midhuna and Dimri, 2019; Hunt and Zaz, 2022), El Niño–Southern Oscillation (ENSO) (Yadav et al., 2010; Dimri, 2013; Kar and Rana, 2014; Cannon et al., 2017; Kamil et al., 2019; Rana et al., 2019; Bharati et al., 2024), Polar/Eurasian Pattern and Siberian High (Wu and Wang, 2002; Cannon et al., 2014), Madden–Julian Oscillation (Barlow et al., 2005; Cannon et al., 2017) and Indian Ocean Dipole (Yadav et al., 2007; Hoell et al., 2013) on intraseasonal and interannual timescales. In particular, the ENSO exerts the strongest influence on the interannual variability of winter precipitation in KH (Rana et al., 2019). One of the key aspects of ENSO teleconnection to Indian Himalayas is the southward shift in the latitude of the winter STJ over the KH during the positive phase of ENSO (Cannon et al., 2014, 2017), which leads to heavier WD precipitation as their tracks move closer to their primary moisture source, the Arabian Sea (Bharati et al., 2024).

Precipitation gauges in the Himalayas are sparse and recognised as inadequate for accurately measuring snowfall (Anders et al., 2006; Rana et al., 2015). While satellite records of precipitation are available, they cover only a limited time frame, whereas our study requires long-term data to analyze the interdecadal variability of precipitation over the KH region. We currently have an 85-year-long reanalysis from ERA5, which has demonstrated a high degree of similarity in both the





quantity and variability of winter precipitation across all time scales when compared to observations
and satellite data in the KH region (Baudouin et al., 2020). The long dataset from ERA5 is
sufficient to examine the interdecadal variability of DJF snowfall over KH. The low-frequency
modes of atmospheric variability such as the Pacific Decadal Oscillation (PDO), Inter-decadal
Pacific Oscillation (Mantua et al., 1998; Zhang et al., 1997; Power et al., 1999; Dai, 2013), and the
Atlantic Multi-decadal Oscillation (Enfield et al., 2001) are known to modulate the regional climate
of the Northern Hemisphere over inter-decadal to multi-decadal timescales. Among these, the PDO
is the dominant mode of SST oscillation in the North Pacific, influencing long-term precipitation
patterns globally (Dettinger et al., 1998; Krishnamurthy, 2013, 2014; Wang et al., 2014; Dong and
Dai, 2015; Yang et al., 2017; Wu and Mao, 2016; Qin et al., 2017). For example, Indian monsoon
rainfall and autumn precipitation in North Central China were found to show an inverse relationship
with PDO (Krishnan and Sugi, 2003; Krishnamurthy, 2014; Qin et al., 2017). However, there is a
major gap in understanding how the PDO affects precipitation over any parts of Himalayas during
any of the seasons.

The current study aims to address this knowledge gap by examining the modulation of the
interdecadal variability of winter snowfall over KH by PDO. Our study aims to understand the
potential influence of the PDO on the Karakoram anomaly, which deviates from the general climate
change patterns observed in the KH region and other mountainous areas. The main objective of this
study are: (1) To examine the spatial distribution of decadal snowfall in KH in different phases of
PDO, (2) how the PDO adjusts global circulation patterns, leading to changes in the STJ, and (3)
how these changes cause impact on a local scale over the KH through WDs and moisture transport.

**2) Data and Methods:**
**2.1 Data**
**2.1.1) Meteorological data**
The study uses meteorological data including geopotential height, zonal (u) and meridional wind (v)
at 200 hPa level, vertically averaged temperature from 500 to 300 hPa level, vertically integrated
moisture flux (VIMF), vertically integrated moisture flux convergence (VIMFC), and global sea
surface temperature (SST) obtained from the European Centre for Medium-Range Weather
Forecasts (ECMWF) ERA5 reanalysis from 1940 to 2022. The jet latitude and strength are





computed by 200 hPa zonal winds over the region (50º – 80ºE, 10º – 60ºN). The jet latitude is the
mean of the latitudes with the largest value of u for each longitude and jet strength is the mean
value of u along these latitudes. ERA5 data have global coverage at hourly frequency and a
horizontal resolution of 0.25°.

**2.1.2) Precipitation data**
Precipitation in the KH is mainly observed through satellite derived and reanalysis products
(Bosilovich et al., 2008; Joshi et al., 2012; Ménégoz et al., 2013; Palazzi et al., 2013; Rana et al.,
2015; Kishore et al., 2016; Baudouin et al., 2020) due to limited and unreliable observations from
ground stations in this complex topographical region (Anders et al., 2006; Bookhagen and Burbank
2006; Strangeways, 2010; Rana et al., 2015; Dahri et al., 2018). The ERA5 reanalysis has
frequently been used for precipitation and snow in recent studies over the KH (Dahri et al., 2018;
Baudouin et al., 2020; T. Singh et al., 2021) and neighbouring mountainous areas (Hu and Yuan,
2020; Li et al. 2021; Dollan et al., 2014). ERA5 closely matches the most reliable gridded
measurements over KH in terms of amount, seasonality, and variability across all timescales during
winter (Baudouin et al., 2020). However, the accuracy of precipitation datasets varies depending on
the season in the region.
To assess the performance of ERA5 precipitation, we compared the ERA5 precipitation with
various gridded precipitation datasets over the KH, including reanalysis datasets from ECMWF
ERA5-land, Modern Era Retrospective-analysis for Research, Applications version 2 (MERRA2),
and High Asia Refined analysis version 2 (HAR v2), as well as rain gauge, and satellite data from
Climate Research Unit version 7 (CRU_TS v7), Global Precipitation Climatology Center version
2022 (GPCC), Global Precipitation Climatology Project version 3.2 (GPCP v3.2), Asian
Precipitation - Highly-Resolved Observed Data Integration Towards Evaluation (APHRODITE
MA_v1101), CPC-Merged Analysis of Precipitation (CMAP), Tropical Rainfall Measuring Mission
(TRMM) Multi-satellite Precipitation Analysis (TMPA) 3B43, and Global Precipitation
Measurement mission-Integrated Multi-satellite Retrievals version 7 (GPM_IMERG v7).
We calculated the linear correlation coefficient between area-averaged precipitation over the KH in
ERA5 and that in other selected datasets. A high correlation was found between DJF ERA5
precipitation and other precipitation products such as APHRODITE and GPCC (Table.1). All the
reanalysis products, including ERA5, demonstrate similar winter precipitation quantities and
variability as found in observations over the KH region across all timescales (Baudouin et al. 2020).



As all of DJF precipitation in KH is in the form of snowfall (fig. 1b), we use ERA5 snowfall data to
investigate the decadal variations of snowfall in the KH (73º – 78ºE, 33º – 38º N).

**Table:1 Correlation coefficents of DJF precipitation based on reanalysis, rain-gauge and**
**satellite with ERA5 precipitation**

|  | Name | Time | Spatial resolution | Correlation with ERA5 | Source |
|---|---|---|---|---|---|
| **Reanalysis** | ERA5-land | 1980-2023 | 0.25° | 0.99 | Hersbach et al., 2018 |
|  | HAR v2 | 1980-2020 | 0.1° | 0.97 | Wang et al., 2021 |
|  | MERRA2 | 1980-2023 | 0.5° | 0.92 | Gelaro et al., 2017 |
| **Rain-gauge based** | CRU_TS v7 | 1980-2023 | 0.5° | 0.84 | Harris et al., 2014 |
|  | GPCC v2022 | 1980-2020 | 2.5° | 0.82 | Schneider et al., 2018 |
|  | GPCP | 1998-2023 | 2.5° | 0.80 | Adler et al., 2016 |
|  | CMAP | 1980-2023 | 2.5° | 0.43 | Xie and Arkin, 1997 |
|  | APHRODITE | 1951-2007 | 0.25° | 0.77 | Yatagai et al., 2012 |
| **Satellite** | GPM_IMERG v07 | 2000-2023 | 0.1° | 0.82 | Huffman et al., 2015 |
|  | TRMM 3B43 | 1998-2019 | 0.25° | 0.80 | Huffman et al., 2007 |


**2.1.2) PDO index**




The PDO index from the National Oceanic and Atmospheric Administration National Climate Data
Center (NOAA-NDC) (https://www.ncei.noaa.gov/access/monitoring/pdo/) is employed to describe
the interdecadal variability of the Pacific Ocean over the period 1940 to 2022.

**2.1.3) Western disturbance data**
WD statistics are computed from the WD track catalogue described in Hunt et al., (2018) and
Nischal et al., (2022), which is based on ERA5 reanalysis data that is spectrally truncated to T42 to
remove noise and small-scale structures. The tracking algorithm detects WDs by identifying upper-
tropospheric regions of positive relative vorticity averaged between 450 hPa and 300 hPa, with the
locations of candidate WDs identified as centroids of these regions. The candidate WDs are then
further refined by only accepting those: 1) whose locations are linked through time to form tracks
that generally follow the westerly steering winds associated with the STJ, 2) that persist for at least
48 hours, and 3) that pass through north India (50°–77°E, 22°–42.5°N). The northern limit of this
box, 42.5ºN, is more poleward than has been used previous studies (36.5ºN). This allows us to
better capture WD impacts over the Karakoram.

**2.2 Methods**

**2.2.1) Lanczos filter**
To isolate the decadal signals, we linearly detrended all meteorological variables and the PDO index
for DJF. These datasets were then filtered using a 9-year running mean Lanczos filter, which is a
low-pass filter based on the sinc convolution (Duchon et al., 1979). The positive (negative) phase of
PDO is defined as years when the filtered DJF PDO index is greater than (less than) zero. We define
the negative epoch (PDO-) as two negative phases of PDO that occurred from 1948 to 1977 and
1989 to 2014, and the positive epoch (PDO+) as a positive phase of PDO that occurred from 1978
to 1988 (fig.1b). Also, the detrended variables are used to conduct correlation and composite
analyses. The Student's and Welch's t-test are used in the study to determine the statistical
significance of correlation and composite analyses, respectively.

**2.2.2) Wavelet analysis**



The PyCWT library (https://pycwt.readthedocs.io/en/latest/tutorial/cwt/) is used to calculate the
cross wavelet power spectrum. This library is based on the implementation by Torrence and Compo
(1998). We employed the cross wavelet transform to calculate the wavelet spectrum between
monthly time series of the PDO index and the area averaged monthly ERA5 snowfall over the KH
region. The cross wavelet transform finds regions in time frequency space where the time series
show high common power.


## 3) Results:

### 3.1) PDO and KH winter snowfall

This study aims to examine the long-term variability in DJF snowfall in the KH region in relation
with the PDO from 1940 to 2022. There is a significant negative correlation between the lowpass-
filtered and detrended time series of DJF PDO and DJF snowfall in the KH (Fig 1b), with a
coefficient of -0.43. However, the PDO is not a single phenomenon, but rather a set of processes
that occur in both the tropics and the extratropics and reflects the influence of various processes
occurring at distinct timescales (Newman et al., 2016). More precisely, elevated sea surface
temperature (SST) in the eastern tropical Pacific is linked to lower SST in the central and western
North Pacific, while higher SST is observed in the eastern North Pacific (Deser et al. 2004;
Newman et al., 2016). Thus, decadal variability of the North Pacific SST can be linked to tropical
Pacific decadal variability, specifically in terms of the long-lasting seasonal ENSO patterns
(Newman et al., 2011; Wittenberg et al. 2014). When the influence of ENSO is eliminated, the
correlation increases slightly to -0.45.

The spatial structure of the correlation between PDO and KH snowfall in winter (Fig 2a) is
significantly negative along the western and central Himalayas and much of the southern
Karakoram, but positive over the Tibetan Plateau and north India. The snowfall in the KH region
during the boreal autumn (SON) and spring (MAM) has a strong positive correlation with the PDO
(not shown), whereas the summer monsoon season (JJA) displays a weak but positive correlation
with the PDO. The different signs of the correlation suggest that the dynamic processes driving KH
snowfall either vary by season, or the seasonal influence of the PDO on KH snowfall changes.



Figure 2b displays the regional distribution of the difference in detrended DJF snowfall between the
negative and positive phases of PDO, hereafter referred to as PDO- and PDO+, respectively. The
difference is significantly positive in the KH area, particularly over the southern part of the
Karakoram region. During PDO+, DJF snowfall over KH is nearly 7% lower than the average
seasonal snowfall, while during PDO- it is about 6% higher. It indicates that the difference in DJF
snowfall in KH varies significantly depending on the phase of the PDO across several decades. DJF
snowfall in the KH accounts for around 80-90% of total annual snowfall during the time period (not
shown), hence a 15% difference in DJF snowfall can have a significant influence on agriculture in
this region, especially since most of the rivers in this region, such as tributaries of Indus, Tarim and
Ganges are partially fed by snowmelt in the spring and later seasons (Armstrong et al., 2018).

This strong relationship between PDO and snowfall in the KH is also demonstrated through a cross
wavelet frequency spectrum analysis between the unfiltered monthly time series of PDO index and
snowfall over the KH from 1940 to 2022 (Fig 2c). The band of strong and significant power in the
period of ~1 year in the cross-wavelet indicates that the PDO and KH snowfall both have strong
interannual variability. The well-known influence of ENSO on snowfall in the region (operating on
interannual timescales) during DJF is also slightly modulated by the low-frequency oscillation of
PDO. Another band of strong power lies in periods of 6-15 years which suggests a robust decadal
scale relationship between these two time series. The significant power in the 6-15-year range
occurred during the periods from 1940 to 1970 and 1998 to 2014, coinciding with the negative
phase of the PDO. There is an insignificant power from 1977 to 1988, which occurred during the
positive phase of the PDO. This suggests that the interdecadal variability of KH snowfall depends
on the phase of the PDO.

**3.2) Sea Surface temperature (SST) variability during DJF**
Figure 3a illustrates the well-known positive (or warm) phase of the PDO over the North Pacific,
shown as a correlation between lowpass filtered and detrended sea surface temperature (SST) and
PDO index during DJF. The correlation pattern also reveals a strong El-Nino like pattern in the
eastern equatorial-tropical Pacific Ocean. For comparison, the correlation pattern between the DJF
SST anomalies and the DJF snowfall anomalies in the KH region is shown in Fig 3b. This
correlation strongly resembles the negative (or cool) phase of the PDO over the North Pacific
Ocean. It is characterised by positive SST anomalies in the northwest Pacific and negative SST





anomalies in the northeast Pacific. Additionally, there are negative SST correlations in the tropical
eastern Pacific region and eastern Indian Ocean adjacent to Western Australia, while positive
correlations are observed in the southwest Indian Ocean and across the northwest Atlantic Ocean.
The correlation pattern in the southern Indian Ocean reveals the subtropical Indian Ocean Dipole
signature (positive phase) (Behera & Yamagata, 2001; Yamagami & Tozuka, 2014).

**3.3) Upper atmosphere circulation response with PDO and snowfall**


In order to understand the anomalous atmospheric circulations that connect the PDO with
anomalous DJF snowfall in the KH region, we computed the correlation of 200 hPa geopotential
height with both the DJF PDO index (Fig 4a) and DJF snowfall (Fig 4b). The correlation pattern
between the PDO and upper level geopotential height shows a prominent upper-level trough over
east China, Japan and the northwest Pacific, which is known as East Asian trough (EAT; Qin et al.,
2018; Yin and Zhang, 2021). In contrast, the correlation pattern over the Caspian Sea, KH, and Lake
Baikal region is associated with positive geopotential height anomalies. The EAT is a well-known
upper atmospheric response to the positive phase of PDO to the East Asia-North Pacific region
during the Northern Hemisphere winter (Newman et al, 2016; Qin et al., 2018; Yin and Zhang,
2021). The intensity of the EAT is strongly linked to the strength of the winter monsoon in East Asia
and the tilt in the EAT axis is connected to midlatitude baroclinic processes, such as the eddy-driven
jet or WD tracks over the East Asia-North Pacific region (Wang et al., 2009). Therefore, changes in
location and intensity of the EAT can lead to, or otherwise indicate, regional climate anomalies,
such as temperature in the upper troposphere which subsequently influence DJF precipitation in
East Asia as well as the KH during the positive phase of the PDO.
These patterns change sign during negative phases of PDO, when KH snowfall is enhanced,
implying an anomalous upper-level trough to the west of the Karakoram, consistent with increased
WD frequency or intensity. The correlation between upper-level geopotential height and snowfall
has a similar pattern to the PDO-geopotential correlation, but as expected, with reversed sign. The
correlation pattern exhibits a strong ridge (or a weakened EAT) over the northwest Pacific and
Japan characterised by the significant positive geopotential height anomalies. The negative
correlation to the west of the KH area shows a trough, which is stronger than the positive
correlation between PDO and geopotential height, indicating the linkage of seasonal snowfall to the
passage of WDs is stronger than the link between the PDO and WDs. Both, however, are important.



The appearance of the anomalous trough in both pairs of correlations implies that the PDO may
affect KH snowfall by somehow modulating WD activity. Therefore, it is essential to understand
how decadal fluctuations in DJF snowfall in the KH are driven by WDs and how the PDO
influences WD behaviour. This can be accomplished by investigating the DJF STJ, followed by a
detailed investigation of the WDs.

**3.4) Modulation of WD and Subtropical Westerly Jet by the PDO**
To further illustrate the above relationship between PDO and DJF snowfall in KH, we examine the
composite differences in 200 hPa wind, geopotential height, and temperature (Fig. 5) between PDO-
and PDO+. Figure 5a displays the difference in 200 hPa circulation over East Asia, Arabian
Peninsula and northwest Pacific region. During the PDO-, there is a large negative geopotential
height anomaly to the north of KH region, which extends from the Caspian Sea-Arabian Peninsula
to KH. Strong westerlies are observed to the south of this trough with a stronger STJ prevailing
across KH during the PDO-. An anomalous trough in the upper atmosphere is indicative of
increased WD frequency (or intensity) and the frequency of WDs is strongly affected by variations
in both the latitude and intensity of the STJ (Dimri et al., 2015; Hunt et al., 2017, 2018) over South
Asia. Therefore we now focus on understanding the relationship between the PDO and the STJ.

Upper-level jets are thermal wind responses to upper-level meridional temperature gradients. In Fig
5b, we show the difference in mid-to-upper (from 500 hPa to 300 hPa) tropospheric temperature
between PDO- and PDO+. A quadrupole in the upper air temperature gradient is present across the
KH, Tibetan Plateau (TP) and the northwest Pacific region during PDO-. Over the Pacific, this is
effectively a direct response to the anomalous surface heating provided by the PDO. Anomalous
warm SSTs over the northwest Pacific lead to adiabatic cooling near the tropopause, which results
in deep convection over the Maritime Continent during the PDO- (e.g., Wang et al., 2016).
Upstream, over continental Asia, the relationship is more complicated and is probably a wave
response to the direct forcing over the ocean. Therefore, a strongly enhanced meridional
temperature gradient over the KH and TP, leading to a stronger and more meridionally-locked STJ.

Figure 5c displays the lowpass filtered time series of latitude and strength of the DJF STJ. During
the PDO-, the STJ tends to sit slightly further north but is also substantially stronger. The





correlation of the time series of the strength of DJF STJ with DJF PDO is significantly negative (-
0.22), and the correlation between DJF STJ strength and DJF snowfall in KH significantly positive
(0.36). The positive (negative) phase of the PDO enhances the movement of the STJ towards the
south (north) through a response to the decreased (increased) SST over the northwest Pacific and
modulates the cyclonic (anticyclonic) circulation over the northwest Pacific and adjacent maritime
continents (Matsumura & Horinouchi, 2016). During PDO-, we observed a quadrupole in the
anomalous upper level temperature gradient (Fig. 5b), resulting in a negative anomaly in the
temperature gradient and an anticyclonic circulation (Fig. 5a) over the TP. Thus, by modulating the
STJ, the negative phase of the PDO leads to more frequent (more intense) WDs at slightly higher
latitudes than usual (e.g. into the Karakoram, where the signal is the strongest).

The presence of a stronger STJ along with a wave-like pattern of trough (anomalous cyclone) over
the northern region of KH, and a ridge (anomalous TP anticyclone) in the upper atmosphere,
increases the occurrence of WDs over KH during the PDO-. After examining the impact of the PDO
on the STJ, we now quantify its influence on WDs directly. Maps of the difference in the frequency
of DJF WDs between PDO- and PDO+ (Fig 6a) indicate that WDs are more frequent (with a 9%
higher frequency) over the KH region during PDO- compared to PDO+. Also, the frequency of
WDs is found to be reduced by around 3% in both the northern and southern regions of the KH
during PDO- compared to PDO+. These WDs are observed to be more intense in the vicinity of the
Caspian Sea and north of the KH during PDO- rather than PDO+ (not shown).

**3.5) Atmospheric-ocean response of PDO on moisture transport in KH**
Increased frequency and intensity of WDs have a significant impact on precipitation in the KH and
surrounding region because they govern southwesterly moisture transport from the Arabian Sea
(Baudouin et al., 2021; Hunt and Dimri 2021). The composite difference of DJF VIMF and VIMFC
between PDO- and PDO+ is now examined to determine the response of moisture transport to the
PDO and its subsequent effect on the KH (Fig. 6b)
The average difference of VIMFC between PDO- and PDO+ is about $0.8 \times 10^{-5}$ kg m$^{-2}$ s$^{-1}$ within
KH region. An advection of moisture from the Black Sea, Red Sea, and eastern Mediterranean Sea
through the Arabian Peninsula/Arabian Sea towards the KH in westerly fashion is observed. The
precipitation associated with WDs is mostly determined by their intensity and proximity to the
Arabian Sea (Baudouin et al., 2020). The variations in the moisture transport across the Arabian





Peninsula/Arabian Sea are not directly linked to changes in VIMF over the northwest Pacific, but the presence of more WDs south of the strong DJF STJ over KH clearly result in greater moisture transport towards KH during PDO-. Hence, the anomalous moisture transport nearly perpendicular to KH, results in increased moisture flux convergence and consequently greater precipitation in the region during PDO-.

## 4) Conclusion and Discussion:

The recemt impacts of climate change over the KH, particularly in mean and extreme winter precipitation, have been largely attributed largely to anthropogenic forcing, such as greenhouse gases, aerosols, and changes in land use. However, these changes cannot be solely explained by natural forcing (Krishnan et al., 2018). Oceanic conditions, especially changes in SSTs over the equatorial-tropical Pacific and north Pacific, play an important role in driving interdecadal variability in atmospheric circulation and hence winter precipitation over the KH .

Understanding this interdecadal variability and its relationship with the PDO is important for understanding the long-term climate of the KH. We have analysed the long-term variability in winter snowfall over the KH due to the PDO by using ERA5 reanalysis data from 1940 to 2022. We found that a strong negative correlation of -0.43 between the PDO and DJF snowfall in the KH. Mean KH snowfall during DJF is approximately 6% greater than the DJF seasonal average during PDO-, and 7% lower during PDO+.

PDO associated anomalous warming of SST in the northwest Pacific modulates the snowfall in the KH via changes in upper-level temperatures over the Pacific and Asia. The warm SSTs lead to increased deep convection and subsequent upper-tropospheric adiabatic cooling over the Pacific. During PDO-, the anomalous heating of the tropospheric column over North Pacific leads to a wave like pattern with an upper-level trough over the north of KH and upper-level ridge over the Tibetan Plateau. This results in a stronger STJ to the west of, and over, the KH, before it is deflected northwards over the Tibetan Plateau. There is a strong positive correlation between the strength of DJF STJ and DJF snowfall in KH, with a correlation coefficient of 0.36, and a significant negative correlation between the strength of STJ and PDO, with a correlation coefficient of -0.22 during DJF at decadal scale. These results suggest a wave response over KH to the direct forcing over the Pacific Ocean.





These anomalous jet conditions over KH are linked to a higher occurrence of WDs across the
region. Using a track catalogue, we found that WDs are 9% more frequent across the KH and drop
by approximately 3% in both the northern and southern regions of the KH during PDO- compared
to PDO+. However, the WDs are found to be more intense in the vicinity of the Caspian Sea and
north of the KH during PDO- rather than PDO+, which is not shown in this study. This increase in
WD frequency results in anomalous moisture transport from the Arabian Sea, Black Sea, Red Sea,
and eastern Mediterranean Sea towards the KH. The moisture transport is almost perpendicular to
the orography of the KH, leading to a strong moisture convergence and thus increased DJF
precipitation in the region during the negative phases of the PDO.
Our findings highlight the importance of considering interdecadal variability when trying to
quantify the effects of anthropogenic climate change in the KH. The recent PDO- has led to
increased WD activity, and hence increased winter snowfall over this region, and may be masking
the effects of climate change. More research is needed to disentangle climate change from the
effects of interdecadal variability over this vulnerable region, so that policymakers can be better
informed.

**5) List of figures:**




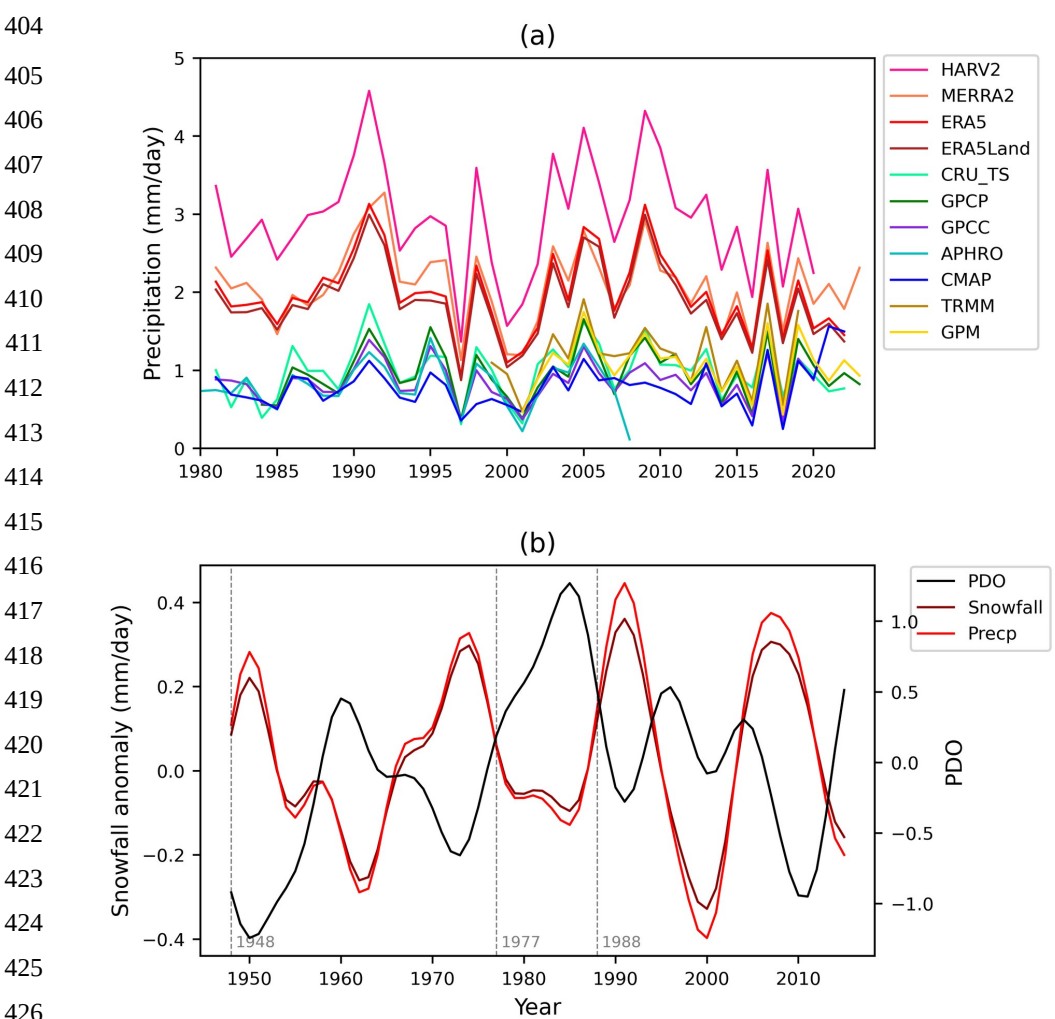

**Figure 1: (a) Seasonal variability of DJF precipitation in KH (green rectangle in fig.2; 73-78E, 33-38N) from ERA5, ERA5-land, MERRA2, HARv2, CRU_TS, GPCP, GPCC, and CMAP during the period from 1980 to 2020, APHRODITE from 1951 to 2007, TRMM from 1998 to 2019, and GPM from 2000 to 2023. (b) Time series of 9-year filtered DJF PDO index and area-averaged DJF ERA5 snowfall (and precipitation) anomalies over KH from 1940 to 2022. The vertical grey lines represent phase transitions of PDO.**



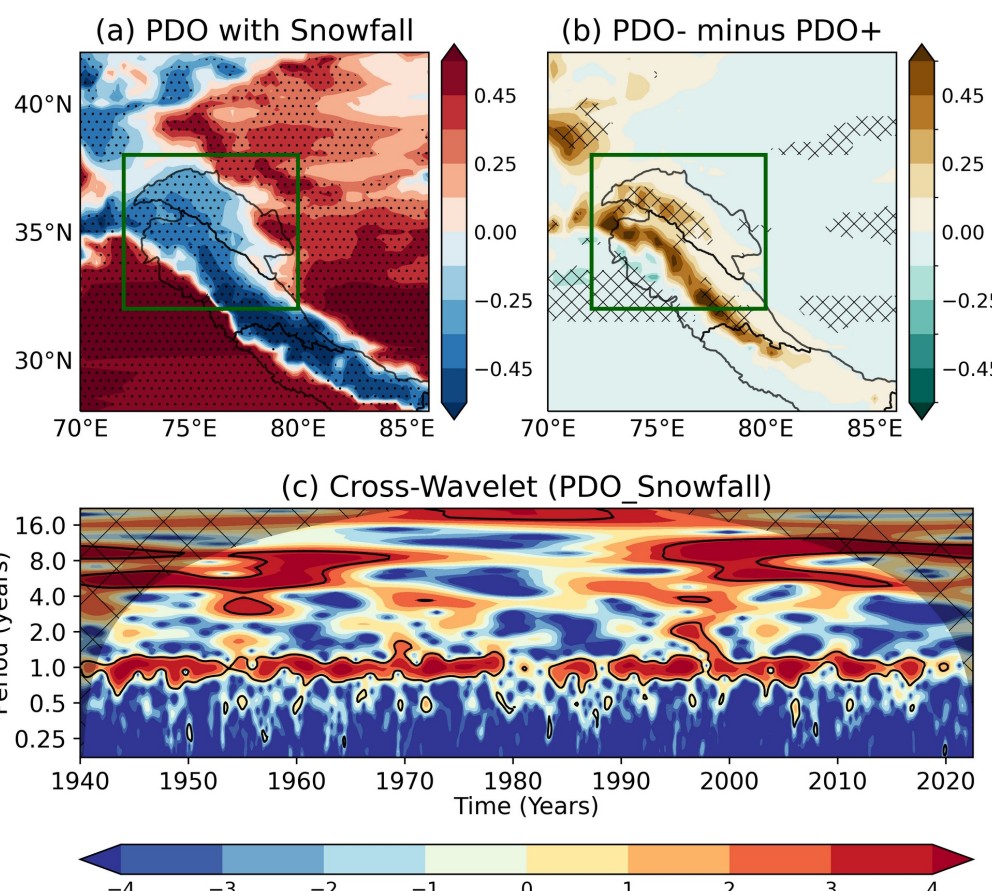

**Figure 2: (a) Spatial map of correlation between the 9-year filtered PDO index and the snowfall over KH during DJF, and (b) composite difference of DJF snowfall between negative and positive epoch of PDO, (c) cross-wavelet of DJF snowfall over KH and DJF PDO index from 1940 to 2022. Stippling in (a) and (b) indicate where the correlation, composite differences are significant at a 95% confidence level. Black line contours on the power spectra in (c) indicate where the spectral power of the cross-wavelet is significantly greater than zero at a 95% confidence level.**








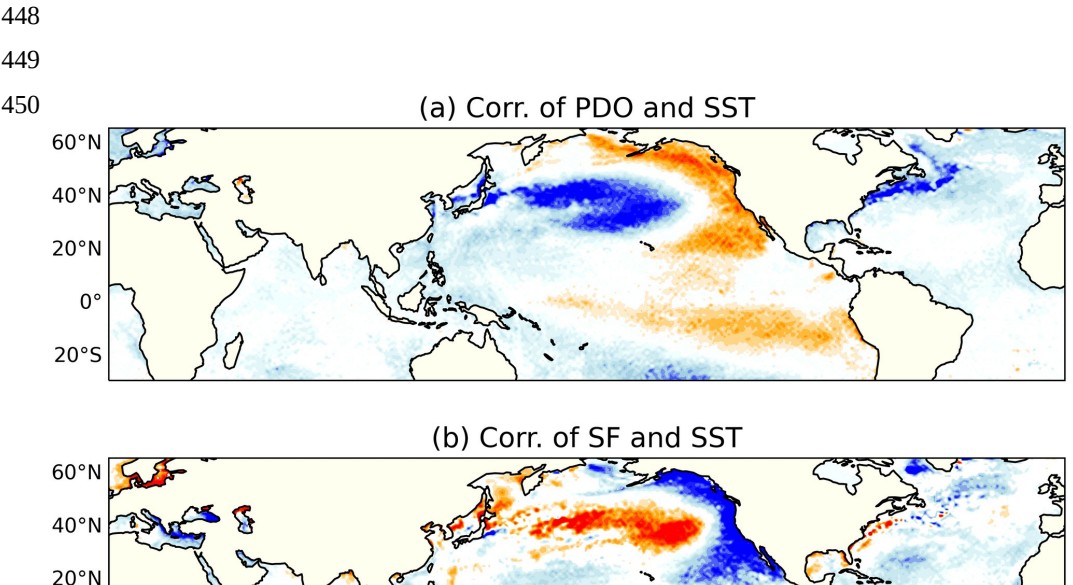

**Figure 3: Spatial map of correlation of the 9-year filtered (a) DJF PDO index, and (b) area**
**averaged DJF snowfall over the green box (fig.2) with 9-year filtered DJF sea surface**
**temperature from 1940 to 2022. The correlations are significant at a 95% confidence level.**













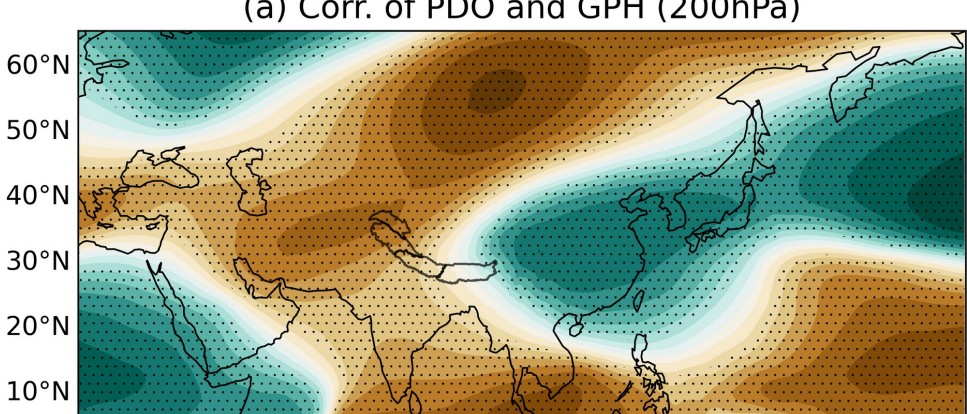

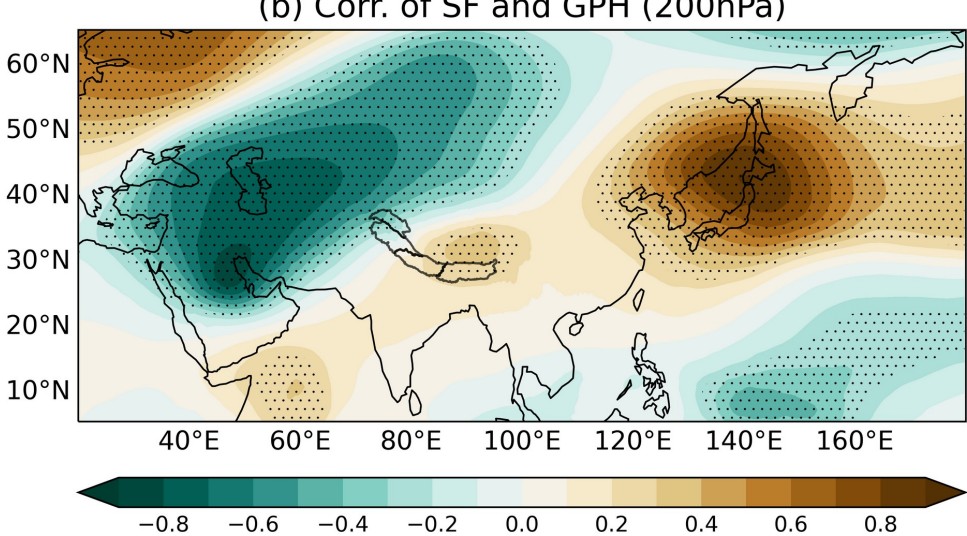

**Figure 4: Spatial map of correlation of the 9-year filtered (a) DJF PDO index, and (b) area averaged DJF snowfall over the green box (fig.2) with 9-year filtered DJF geopotential height at 200hPa (m) from 1940 to 2022. Stipplings in (a) and (b) indicate where the correlations are significant at a 95% confidence level.**









**Figure 5: Composite difference of (a) U-wind (colour; m/s), wind (vectors; m/s), and**
**geopotential height (contours; m), (b) vertically averaged temperature (C) from 300hPa to**
**500hPa level during DJF between negative and positive epoch of PDO, (c) time series of 9-year**
**filtered strength (red) and latitude (blue) of DJF subtropical westerly jet (STJ) over KH**
**(green box; fig.2) from 1940 to 2022.**


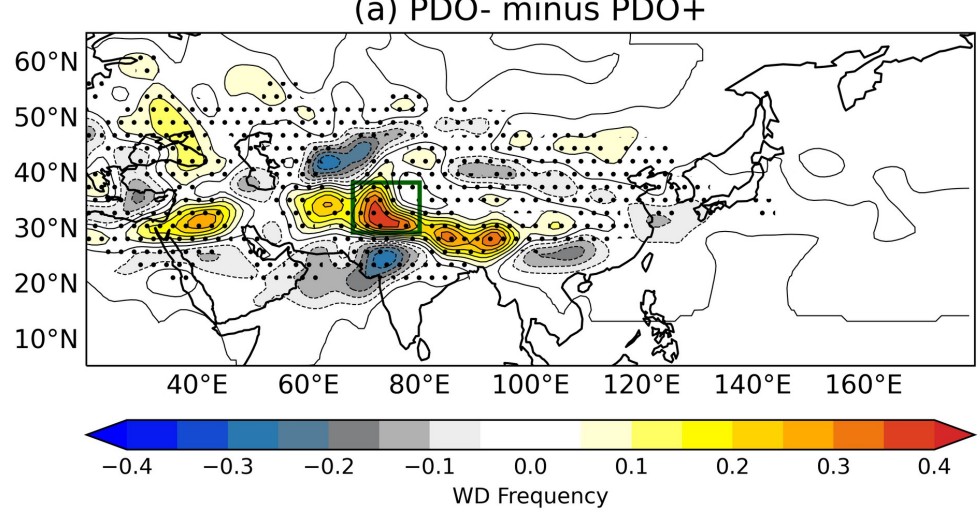

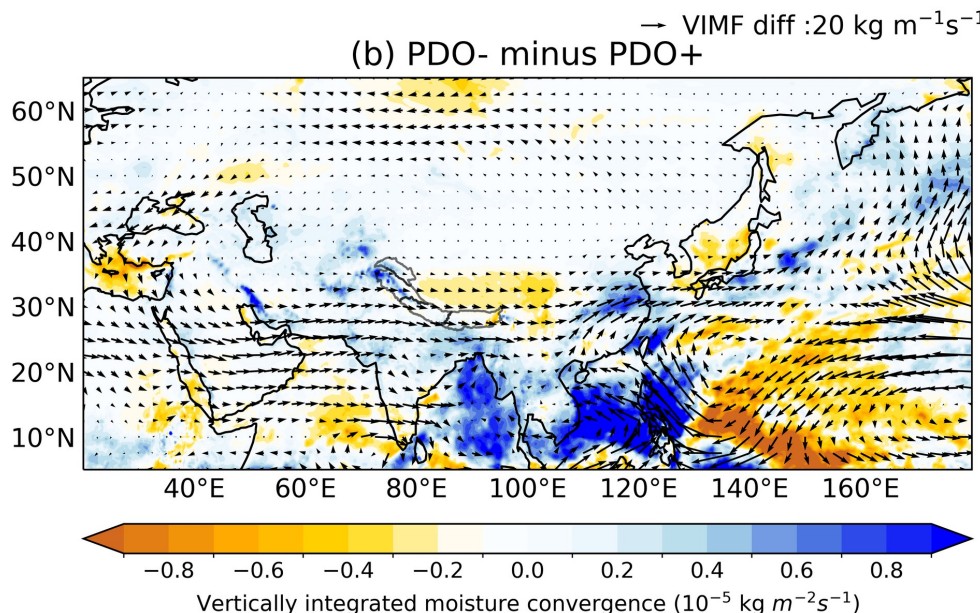

**Figure 6: Composite difference of (a) WD frequency, and (b) vertically integrated moisture**
**flux (vectors; kg/m/s) and vertically integrated moisture convergence (colours; kg/m2/s)**



**during DJF between negative and positive epoch of PDO from 1940 to 2022. Stippling in (a)**
**indicates where the differences are significant at a 95% confidence level.**

## Author Contributions:

**Priya Bharati:** conceptualization; formal analysis; methodology; investigation; software;
visualization; writing original draft. **Kieran M. R. Hunt:** conceptualization; methodology;
software; writing - review and editing. **Pranab Deb:** supervision; conceptualization; writing –
review and editing.

**Competing interests:** The contact author has declared that none of the authors has any
competing interests.

## Acknowledgements:

The work is carried at CORAL, Indian Institute of Technology Kharagpur under supervision of
Pranab Deb. Priya Bharati is funded through the Ministry of Science and Technology, Government
of India, Council of Scientific and Industrial Research (CSIR; 09/081(1371)/2019-EMR-I). KMRH
is supported by a NERC Independent Research Fellowship (MITRE; NE/W007924/1).

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
