# Peer review of "PDO-driven interdecadal variability of snowfall over the Karakoram and Western Himalaya"

_EGUsphere, 2024_

## Referee Comment (RC1)

**Review of "PDO-driven interdecadal variability of snowfall over the Karakoram and Western Himalaya" by Bharati, Deb and Hunt**

*Submitted to WCD*

This manuscript explores the relationship between the Pacific decadal oscillation (PDO) and snowfall over the Karakoram and Western Himalaya during the period 1940-2022 using the ERA-5 reanalysis. Understanding the drivers of snowfall variability in the region is import. In particular, could snowfall trends explain the "Karakoram anomaly" where glaciers in the Karakoram region have been stable unlike most other mountain glaciers globally in large part due to winter snowfall preserving the glacier mass balance. Western Disturbances (WD) contribute significantly to winter snowfall in this region, and so PDO impacts on WDs provides a mechanism by which PDO can drive snowfall variability. This is a difficult problem to study as there are few long-term snowfall observations in the region and both satellite retrievals and reanalysis show biases over the mountainous terrain of the Karakoram and Western Himalaya. Long datasets are required to study decadal oscillations such as the PDO. The result is that I am not completely confident in the robustness of the results, and this at least needs mentioning in the discussion section in the manuscript.

Comments:

1) As noted, it is hard to get good long-term observations of snow fall, but figure 1 shows that the reanalysis datasets consistently overpredict precipitation compared to the satellite datasets (though which is "correct" is not completely clear). In terms of variability then all the datasets (with the possible exception of CMAP) are reasonably well correlated with ERA5 (although this is DJF precipitation averaged over quite a large box so not the toughest of tests). Maybe this correlation is enough if you are looking at how snowfall variability is modulated with the PDO, but it would be good to explicitly discuss this.

2) Why did you choose the green box in figure 2 for averaging over? I can appreciate the simplicity of using a box, but it includes regions where snowfall is negatively correlated with PDO over the mountains, while it is positively correlated with PDO in both the foothills to the SW in the box and in the NE of the box. This potentially complicates the interpretation of later results.

3) What are the thin black lines on Figure 2 (and other figures). State boundaries? These are helpful in terms of comparing figures, so it would be good to explain and refer to them in the text.

4) Is the record of 1940-2022 long enough? The record only contains two periods of negative PDO and one short (11 year) period of positive PDO. Comparing PDO+ and PDO- periods is therefore not very robust. I am not sure what else you can do given data availability (perhaps 20[th] century reanalysis project?), but at the very least this limitation needs discussing.

5) In general, there is some blurring between results and discussion (e.g. lines 234-236 feel more like discussion not results).

6) Figure 2b – what is shown here? If this is a difference in snowfall, then is should presumably be a depth with units of length? Please clarify and include the proper units.

7) The interpretation of figure 2c (lines 248-249) talks about the interdecadal variability of KH snowfall depending on the PDO phase. I am not sure you can say that from the results. The power in the 6-15 year band is only significant in the PDO- phase, and so what the results suggest is that in the PDO- phase PDO and KH snowfall are related on a decadal timescale,

but again this is only over a 30 / 16 year time frame for the 1$^{st}$ and 2$^{nd}$ PDO- periods and so it is not clear that this is significant.

8) Figure 6b. The additional moisture convergence is relatively modest over the region where there is significant additional snowfall (figure 1b), so how can you be confident this is driving the additional snowfall? Much larger convergence is seen elsewhere.

9) It would be good to mention the uncertainties of the research in the discussion and conclusion, for example the uncertainties in the snowfall datasets and the challenges of a relatively short timeseries for studying decadal oscillations. If models capture the coupling between PDO and snowfall then potentially large ensembles or longer climate simulations could be used to confirm these conclusions? I'm not suggesting you need to do this, but it would be worth mentioning as future work.

---

## Author Comment (AC1)

**Review of "PDO-driven interdecadal variability of snowfall over the Karakoram and Western Himalaya" by Bharati, Deb and Hunt**
**Submitted to WCD**

**Comments:**

1) As noted, it is hard to get good long-term observations of snowfall, but figure 1 shows that the reanalysis datasets consistently overpredict precipitation compared to the satellite datasets (though which is "correct" is not completely clear). In terms of variability then all the datasets (with the possible exception of CMAP) are reasonably well correlated with ERA5 (although this is DJF precipitation averaged over quite a large box so not the toughest of tests). Maybe this correlation is enough if you are looking at how snowfall variability is modulated with the PDO, but it would be good to explicitly discuss this.

*Reply: We agree with the challenges associated with long-term snowfall monitoring in this complex topographical area. The reanalysis datasets overestimate precipitation relative to satellite datasets and some rain-gauge datasets; yet, the seasonal variability of precipitation exhibits comparable variability among all datasets in relation to ERA5. Also as the reviewer points out, the correlation coefficients are very high between the reanalysis and most satellite/gauge-based products. The primary reason for choosing ERA5 is its longevity – none of the other datasets have long enough observation periods for a decadal study. Also, although ERA5 precipitation exhibits biases in its quantity throughout this region, it has been used in other studies for analysing seasonal snowfall variability. We have added some text discussing this in our revised manuscript.*

2) Why did you choose the green box in figure 2 for averaging over? I can appreciate the simplicity of using a box, but it includes regions where snowfall is negatively correlated with PDO over the mountains, while it is positively correlated with PDO in both the foothills to the SW in the box and in the NE of the box. This potentially complicates the interpretation of later results.

*Reply: : Thank you for this useful suggestion. We now use a KH shapefile to select the Karakoram and Western Himalaya regions only. This has improved the strength of correlations presented in the study, and simplified the interpretation of the results. The revised plot in Figure 2 is shown below and discussions have made in the lines from 218 to 226 in revised manuscript.*

[Figure]

(a) PDO with Snowfall

(b) PDO- minus PDO+

(c) Cross-Wavelet (PDO_Snowfall)

3) What are the thin black lines on Figure 2 (and other figures). State boundaries? These are helpful in terms of comparing figures, so it would be good to explain and refer to them in the text.

*Reply: No, these lines demarcate the traditional boundaries of the Karakoram, Western, and Central Himalaya. We now mentioned this in the revised caption of Figure 2.*

4) Is the record of 1940-2022 long enough? The record only contains two periods of negative PDO and one short (11 year) period of positive PDO. Comparing PDO+ and PDO- periods is therefore not very robust. I am not sure what else you can do given data availability (perhaps 20th century reanalysis project?), but at the very least this limitation needs discussing.

*Reply: We agree with the reviewer. The lack of availability of long-term observations and reanalysis products of precipitation in the KH region complicates the understanding of low-*

*frequency precipitation variability in this area. However, the ERA5 dataset is accessible for over 80 years, which can be utilized to illustrate the long-term variability in precipitation within the region influenced by decadal teleconnections. We accept that the occurrences of PDO phase changes are limited in number; however, the duration of these periods is sufficiently extensive to demonstrate the impact of PDO phase changes on precipitation. We also use correlation-based analysis to strengthen our arguments. We have revised the manuscript to discuss this limitation and note that the same type of analysis can be done in future work with the help of longer reanalyses or global modelling experiments.*

5) In general, there is some blurring between results and discussion (e.g. lines 234-236 feel more like discussion not results).

*Reply: We have moved these sentences to the discussion in our revised version.*

6) Figure 2b – what is shown here? If this is a difference in snowfall, then is should presumably be a depth with units of length? Please clarify and include the proper units.

*Reply: The unit (mm day$^{-1}$) has been added in the plot and its caption in figure2b.*

7) The interpretation of figure 2c (lines 248-249) talks about the interdecadal variability of KH snowfall depending on the PDO phase. I am not sure you can say that from the results. The power in the 6-15 year band is only significant in the PDO- phase, and so what the results suggest is that in the PDO- phase PDO and KH snowfall are related on a decadal time scale, but again this is only over a 30 / 16 year time frame for the 1st and 2nd PDO- periods and so it is not clear that this is significant.

*Reply: The correlation between PDO phase shifts and winter snowfall in the KH has been updated in the revised manuscript following the reviewer's suggestions of the new bounding box (see the comment 2 and plot) This leads to larger areas of significance in the wavelet analysis. The significant power in the 6-15-year range occurred between 1940 and 1970 and again from 1998 to 2015, coinciding with the negative phases of the PDO. An insignificant weak power appeared within the same range from 1971 to 1988, coinciding with the positive phase of the PDO. A long band of strong power exists throughout the 16–20-year range, observed from 1950 to 1990, while a weaker power is shown from 2000 to 2022. This indicates that the low-frequency variability of KH snowfall is influenced by decadal oscillations over various time scales, while the interdecadal variability of KH snowfall is found to influenced by the phase of the PDO.*

8) Figure 6b. The additional moisture convergence is relatively modest over the region where there is significant additional snowfall (figure 1b), so how can you be confident this is driving the additional snowfall? Much larger convergence is seen elsewhere.

*Reply: Moisture convergence is 16% greater during the negative phase of the PDO compared to the positive phase. This fractional change is statistically significant and leads a fractional increase in snowfall over the KH of similar magnitude. Absolute values of moisture convergence are greater in other areas due to greater atmospheric moisture content.*

9) It would be good to mention the uncertainties of the research in the discussion and conclusion, for example the uncertainties in the snowfall datasets and the challenges of a relatively short timeseries for studying decadal oscillations. If models capture the coupling between PDO and snowfall then potentially large ensembles or longer climate simulations could be used to confirm these conclusions? I'm not suggesting you need to do this, but it would be worth mentioning as future work.

*Reply: This is a highly beneficial suggestion for the paper. The proposed uncertainties and gaps are indicated as highlighted text in the revised version in lines from 408 to 417.*

---

## Author Comment (AC2)

**Review of the manuscript egusphere-2024-2845 entitled "PDO-driven interdecadal variability of snowfall over the Karakoram and Western Himalaya" by Bharati et al.**

Snowfall not only affects the mass balance of glaciers but is also an important source of water resources in arid and semi-arid areas. The Karakoram and Western Himalaya regions are home to numerous high mountain glaciers, serving as the sources for rivers such as the Indus, Tarim, and Amu Darya. The unique characteristics of the Karakoram anomaly have drawn sustained attention from the academic community. This article explores the relationship between the Pacific Decadal Oscillation (PDO) and winter snowfall in the Karakoram-Western Himalaya region, analysing the mechanisms involved. The paper is well-written and presents a clear line of thought. I recommend that the paper be published in Weather and Climate Dynamics following revisions.

**#Major**

1) Lines 98-104: While these interdecadal factors (PDO, IPO, and AMO) significantly influence global climate, could you explain why this article focuses solely on the impact of the PDO?

*Reply: Our decision to investigate the role of the PDO follows from Fig 15 in the recent review of WDs by Hunt et al (2024; doi:10.5194/egusphere-2024-820). There, they perform a correlation between WD activity and SSTs, and highlight the potentially important role of both the PDO and NAO. Similarly, our own Figure 3, where we correlate decadal-filtered SSTs with decadal-filtered WH winter snowfall, shows no signal in either the AMO or IPO. For these reasons, we have focused on the PDO. We have also compute the partial correlation between snowfall in WH and PDO after excluding the ENSO, IPO and AMO influence (explanation mentioned in lines from 220 to 226) in the revised manuscript.*

2) Lines 107-109: A published study examines the impact of the PDO on non-monsoon season precipitation in the northwestern Himalayas, highlighting its role in regulating westerly disturbances (Aggarwal et al. 2024, CD). Please introduce it here and emphasize the differences with this work.

*Reply: The suggested point has mentioned in the revised manuscript.*

3) From Figure 1a, it can be observed that the precipitation values from the reanalysis data are significantly higher than those from the assimilated grid precipitation data. Additionally, the reanalysis data exhibits more pronounced interdecadal variability. What factors contribute to this discrepancy? This article is primarily based on ERA5 data. what impact does this have on the results? Please discuss.

*Reply: We agree with the challenges associated with long-term snowfall monitoring in this complex topographical area. The reanalysis datasets overestimate precipitation relative to satellite datasets and some rain-gauge datasets; yet, the seasonal variability of precipitation exhibits comparable variability among all datasets in relation to ERA5. Also, the correlation coefficients between the reanalysis and most satellite and observational datasets is very high as we state in Table 1, implying a very high covariance. Although ERA5 precipitation exhibits biases in its quantity throughout this*

*region, it has been used in other existing study for analysing seasonal snowfall variability. Our choice of ERA5 thus has almost no effect on the results, except for its longevity.*

*Following minor point (5) of this reviewer and major point (2) of the other reviewer, we have replaced our simple box-based regional selection with a more accurate boundary for the KH. This further improves the correlation of all precipitation datasets with ERA5 precipitation over the KH.*

4) From your analysis, it is clear that there is a significant correlation between PDO, WD, and snowfall. However, the article does not clearly explain how the PDO influences the westerly disturbances in the Karakoram-Western Himalayan region through teleconnection processes. Additionally, is it possible to validate the process by which the PDO affects winter snowfall in the Karakoram-Western Himalayan region through modelling experiments?

*Reply: We disagree with the reviewer. We have shown that the negative phase of the PDO leads to more frequent (more intense) WDs at slightly higher latitudes than usual (e.g. into the Karakoram, where the signal is the strongest) by modulating the STJ. The presence of a stronger STJ along with a wave-like pattern of trough (anomalous cyclone) over the northern region of KH, and a ridge (anomalous TP anticyclone) in the upper atmosphere, increases the occurrence of WDs over KH during the PDO-.*

5) Additionally, is it possible to validate the process by which the PDO affects winter snowfall in the Karakoram-Western Himalayan region through modelling experiments?

*Reply: The mechanisms and atmospheric response of the PDO teleconnection to winter snowfall in the Karakoram-Western Himalayas through modelling experiments constitute part of our planned future studies. This is already mentioned in the future work part in the conclusion of this study.*

**#Minor**

1) Line 76: Add a space before WD.

*Reply: Corrected in the revised manuscript.*

2) Please include some recent references in the introduction.

*Reply: The required references have been added in the revised manuscript.*

3) Line 126: The following analysis uses data up to 2023 (Table 1), but here mentions data only up to 2022. Please check.

*Reply: We have done the analysis from 1940 to 2022. The time period is checked for Table.1 in the revised manuscript.*

4) Line 162: Could you clarify the temporal resolution of the data used for the calculations in this section?

*Reply: Monthly temporal resolution is used to analyse all meteorological parameters in this study and updated in the revised manuscript.*

5) Lines 206-217: When calculating the correlation of snowfall in the Karakoram -Western Himalaya region with other indices, was the part of the region with significantly positive correlation coefficients (Fig 2a) excluded? If not, I recommend excluding it before recalculating, as this will likely yield more reliable results.

*Reply: This is very useful suggestion to improve our results. We have computed the correlation after excluding the regions of significantly positive correlation. Our results have been updated accordingly in the revised manuscript. The updated plots in figure 2 are shown in the revised version as following plots:*

[Figure]

6) Line 453: I did not see any information regarding significance testing, please check. If there isn't any, please add it.

*Reply: The plotted SST are only the significant magnitudes from the correlation and significance testing. The detail about testing has updated in the revised manuscript.*

7) Figure 6, please use the same aspect ratio (width to height).

*Reply: Figure 6 is updated in the revised manuscript.*

Related references:

Aggarwal D, Chakraborty R, Attada R. Investigating bi-decadal precipitation changes over the Northwest Himalayas during the pre-monsoon: role of Pacific decadal oscillations. Climate Dynamics. 2024 Feb;62(2):1203-18.

Dimri AP, Pooja, Jeelani G, Mohanty UC. Western disturbances vs Non-western disturbances days winter precipitation. Climate Dynamics. 2023 Nov;61(9):4825-47.

---

## Author Response (AR2)

I noted in the response to point 2) from reviewer 1 that the authors said that they are now using a shape file to average over rather than the green box in the original figure 2. In the track change manuscript however figure 2 and 6 still have the green box on and the text still says that the averaging is done over this box (line 158 and captions to figures 3,4,5). If the new method has been implemented and shows better results, please ensure that the manuscript is updated to reflect this.

***Reply: We modified the specified points in the revised manuscript and highlighted them in the track-change document. The captions for figures 3, 4, and 5, as well as text in line 158, have been modified accordingly.***